# Barriers to Adopting a Plant-Based Diet in High-Income Countries: A Systematic Review

**DOI:** 10.3390/nu16060823

**Published:** 2024-03-14

**Authors:** Alice Rickerby, Rosemary Green

**Affiliations:** Department of Population Health, LSHTM Centre on Climate Change and Planetary Health, London School of Hygiene and Tropical Medicine, Keppel St, London W1CE 7HT, UK; rosemary.green@lshtm.ac.uk

**Keywords:** plant-based diet, barriers, sustainable diets, net-zero, sustainable development goals, perceptions, consumer attitudes, dietary behaviour, COM-B model, behaviour change wheel

## Abstract

Adopting a plant-based diet (PBD) has been shown to reduce the risk of developing certain diseases and is linked to environmental benefits. This review synthesises the evidence on the barriers adults aged 18 to 65 living in high-income countries (HIC) may experience when adopting a PBD. A systematic literature review was conducted using four search databases: Medline, Embase, Global Health, and Web of Science. Barriers were mapped to behaviour change strategies using the COM-B model. Ten studies were included in the final review, including 1740 participants. Five were qualitative, four were cross-sectional, and one was a pre- and-post-intervention study. In total, 40 barriers were identified and synthesised into 11 themes: financial, lack of knowledge, emotional, health, convenience, social, enjoyment of meat, environmental, accessibility, personal ability, and media. Of the 40 barriers, nutritional intake/requirements (categorised into the “health” theme) had the most evidence. This barrier encompassed concerns around being able to meet nutritional needs if an individual were to adopt a PBD. Habits (in the “personal ability” theme), which included established eating habits and habitual behaviours relating to animal-source foods, had the second most evidence alongside the barrier of not knowing what to eat as part of a PBD (in the “lack of knowledge” theme). Education interventions and communication/marketing policies were the behaviour change mechanisms mapped onto these barriers. Future interventions should focus on informing individuals about what to consume as part of a nutritionally balanced PBD and facilitating habitual dietary change.

## 1. Introduction

Dietary changes to reduce consumption of animal-source foods and increase plant-based foods can help meet several sustainable development goals (SDGs) [1]. For example, the production of animal-source protein requires 100 times more water than plant-based sources such as grain proteins [2], so dietary change could have a positive effect on the achievement of SDG6 (ensuring sustainable and accessible sanitised water for all). Land use, deforestation and reduced biodiversity (SDG15), and climate change (SDG13) are also exacerbated by diets high in animal-source foods [3,4], and dietary change could be an effective way to reduce these land use issues. In addition, SDG3 (good health and well-being) and SDG12 (sustainable consumption and production) could all be positively impacted by increasing healthy plant-based foods in diets as a replacement for animal-source foods.

Currently, net-zero pledges are in place to reduce greenhouse gas emissions by 2050 [5]. If all these pledges are realised, they could limit global heating to below the crucial 2-degree target [6], but thus far, there is a large gap between ambition and implementation. This is particularly true in the food and agriculture sector, with no countries yet implementing carbon pricing for the most environmentally damaging foods (largely animal-source foods) and a reluctance to commit to policies to change diets [7]. Although meat consumption has reduced in high-income countries (HICs) such as the UK in recent years, this dietary transition is still not proceeding fast enough to meet net-zero targets [8]. Large reductions in global greenhouse gas emissions as a result of moving away from diets high in animal-source foods have been estimated [9]. This reduction is vital for global sustainability targets since food systems are responsible for a third of all greenhouse gas emissions. Additional environmental pressures include land and water use, threatening worldwide biodiversity [10]. Moving away from diets high in animal-source foods has been shown to reduce these environmental burdens in several studies [9,11]. Plant-based diets can be described as “dietary patterns that have a greater emphasis on foods derived from plants” (such as fruits and vegetables, whole grains, pulses, nuts, seeds, and oils) [12]. Plant-based foods commonly include nutrient-dense legumes, seeds, nuts, whole grains, unsaturated fat, etc. However, they can also include meat and dairy alternatives [12]. In this review, a PBD is defined as predominantly plant-based, such as a flexitarian diet or the “planetary health” diet recommended by the EAT-Lancet Commission [13], rather than exclusively plant-based, such as a vegan diet. The adoption of PBDs has increased in the past few years in high-income countries such as the UK, Australia, the USA, and Israel [14,15,16,17]. For example, in the past ten years, the proportion of adults consuming plant-based alternatives to meat and dairy in the UK almost doubled from 6.7% to 13.1% over ten years [14].

Diets richer in plant-based foods than animal-based foods are associated with a lower risk of developing and dying from cardiovascular diseases, as found in a study conducted in the USA [18]. When comparing different diets, those who consume meat and fish have been shown to have a higher prevalence of hypertension than vegetarians and vegans in the UK [19]. In addition, adherence to a predominantly PBD has been associated with a reduced risk for certain cancers in US studies, including breast cancer [20], prostate cancer [21], and colorectal cancer [22]. These health benefits could be due to an increased dietary fibre intake, as seen in a systematic review from Germany [23], less saturated fat, and lower salt intake, as stated by the WHO [24]. A healthier and more diverse gut microbiome can also be seen in individuals following a PBD. This healthier gut microbiome is due to increased dietary diversity [23] and fibre intake [25]. PBDs tend to be higher in fruit and vegetable content than diets not primarily focused on plant foods. Individuals consuming a vegetarian diet were found to have a higher concentration of antioxidants such as vitamin C, vitamin E, and beta carotene than those consuming an omnivorous diet [26].

Barriers to behaviour change could impede the rate of a dietary transition towards PBDs. These barriers can include issues with accessibility, lack of options for plant-based foods, and financial issues, as seen in a UK study [27], as well as fears of nutritional deficiencies, negative health perceptions [28] from switching to a PBD, and social acceptance in the USA. Some individuals may perceive that plant-based foods are unenjoyable, while others may enjoy meat strongly, as shown in a Swiss study [29]. Furthermore, a lack of dietary information and knowledge on what to eat as part of a PBD may constitute a barrier among individuals without much previous experience with PBDs in their families or social groups, as indicated by an Australian study [30]. In this study, we examine barriers to adopting a PBD and strategies that may help to overcome these using the behaviour change wheel (BCW) created by Michie et al., 2011 [31]. The BCW uses nineteen frameworks and consists of three distinct layers. It uses the capability, opportunity, motivation, and behaviour (COM-B) model in the wheel’ s centre [31]. This model states that for behaviour change to be successful, an interacting system that involves all components must be used, as shown in the wheel. For example, for policymakers to strategically implement new policies, one or more of the components in the BCW must be successful for any influence on behaviour change to occur [32]. A middle layer surrounds the hub, consisting of nine intervention functions, and an outer layer comprises seven policy categories. Both layers can support the influence of each category in the COM-B model.

The COM-B model focuses on the middle section of the BCW. Capability and opportunity influence motivation [33]. These three combined influence behaviour changes. Capability refers to whether an individual has the knowledge, skills, and abilities to engage in certain behaviours. Both psychological and physical capabilities need to be considered for this section. Secondly, opportunity refers to external factors that affect a particular behaviour. This section encompasses the requirement for both physical and social opportunities. Lastly, motivation is the internal driving force that influences behaviours and decision making. Motivation has two components: reflective and automatic motivations.

One study previously conducted a systematic review relating to this topic. The review focused on the facilitators and barriers to adopting a PBD [34], looking particularly at chronic diseases in all income settings and including global studies, with a focus on Hungarian studies. The review defined PBDs as a vegetarian diet and included papers published from 1974 to 2019. Therefore, this systematic review will provide up-to-date evidence using relevant and recent studies focusing on HICs and adults aged 18–65.

## 2. Materials and Methods

### 2.1. Information Sources/Search Strategy

Full ethical approval was obtained from the London School of Hygiene & Tropical Medicine ethics committee before data collection began (Appendix A). The Preferred Reporting Items for Systematic Reviews and Meta-Analysis (PRISMA) was used for this systematic review. This systematic review followed the PRISMA checklist (Appendix A).

In May 2022, a search was conducted over four databases: Medline, Embase, Global Health, and Web of Science. The search strategy included three main concepts: the outcome of a PBD, the geographical setting of HICs, and the study population group of adults aged between 18 and 65. For each concept, a list of keyword search terms was created. These were combined using the Boolean Operators OR and AND. For each database, a separate list of subject headings was also created for those required (Appendix A). Limits and filters were not applied to the search phase; this was conducted after the data were collected during the data screening phase.

### 2.2. Eligibility Criteria

The studies were included or excluded using predefined eligibility criteria:Study population: included human adults aged 18 to 65 years old. Studies were excluded if participants had diabetes, cardiovascular disease, high blood pressure, or obesity.Study setting: studies set in HICs were included, whereas those set in low- and middle-income countries (LMICs) were excluded. These were excluded due to large discrepancies in dietary patterns in middle-income countries. For example, on average, high-income countries consume too much meat and dairy. However, in middle-income countries, there is a much wider variation in dietary patterns. For example, many in Latin America consume high amounts of meat, whilst others (often in Asia) consume less. Therefore, including these countries would have likely introduced too much heterogeneity into the results; however, it would be useful to examine these countries and their different barriers in the future.Exposure: studies that included a perceived barrier to adopting a PBD were included. An example is a financial barrier. Studies that included PBDs were used, and these could be diets considered flexitarian or plant-forward but exclude exclusively plant-based/vegan diets. This type of diet was excluded as evidence on sustainable and healthy diets, including the EAT-Lancet Commission, recommends shifts to diets low in animal-source foods as being attainable for large parts of the population rather than diets where these foods are completely absent.Outcome: due to the nature of this review, defined outcomes were not a necessary part of the inclusion and exclusion criteria.Study design: all study designs were eligible; these included qualitative, mixed methods, and quantitative studies, both observational and experimental.Year: studies conducted before 2000 were excluded; any studies that fit all other eligibility criteria conducted after 2000 were included. The cut-off point for analysis of studies was May 2022.Type of publication: publications that were reviewed and not from the original publication source, such as systematic reviews and grey literature, were excluded.Language: Studies were only included if they were written in English.Studies with a focus on only meat alternatives were excluded.

### 2.3. Screening

Database searches were exported into EndNote 20, where duplicates were removed. Titles and abstracts were screened, and any irrelevant studies were removed. The full texts for each study were obtained and assessed for eligibility (inclusion and exclusion criteria). Reasons for exclusion were documented. Each study was screened and retrieved independently by AR (Appendix A). 

### 2.4. Data Extraction and Analysis

Information was extracted from each eligible study using a data extraction form (Appendix A). This information included authors, title, year of publication, geographical setting, study population sample size, age, sex of participants, study design, outcomes (if present), effect measures (if present), key findings, funding, conflicts of interest, and URLs. This process was completed independently by AR. Defined outcomes were listed in the results table. These included dietary intake and change, public awareness, and perceptions about PBDs. However, only some of the studies fulfilled this criteria. Therefore, non-defined outcomes were also added where necessary. This process was conducted by examining the aims, objectives, and study findings of those studies that did not have clearly defined outcomes. This information created non-defined outcomes. Effect measures were also added to the results table where relevant (Appendix A). However, most studies did not have defined effect measures. Therefore, most of the synthesis did not use quantitative effect measures to show the relationships found, and a meta-analysis was not conducted. Instead, barriers were broken down into themes, and a conceptual map was created to display the synthesised results visually. Links between barriers across the different themes were found and added to the conceptual map. This was an iterative process. The first step of this process was familiarisation with the data. Secondly, a code was attached for each study finding (barrier) derived from identifiable key concepts. These codes were collated together due to concept relevance. These were then placed into overarching themes. These overarching themes were created from patterns identified in the grouped codes. These were reviewed to ensure the themes reflected the overall study findings. The themes were not based on the previous literature as they were devised from the data obtained; they were created from a data-driven process. Furthermore, the conceptual map was created by organising these themes according to a hierarchy concerning their respective codes. This mapping helped form groups and subgroups, which allowed the data to be clearly and concisely presented while forming coherent conclusions and assessing the barriers’ strengths. The results were also mapped against the COM-B model. Each barrier was categorised into the relevant component.

### 2.5. Quality Assessment

For the five qualitative studies, the Critical Appraisal Skills Programme (CASP) for qualitative studies tool was used [35]. The four quantitative studies used the Centre for Evidence-Based Management (CEBMa) critical appraisal tool for cross-sectional studies [36]. Lastly, for the one pre- and post-intervention study, the NICE Graphical Appraisal Tool for Epidemiological Studies (GATE) quantitative intervention checklist was used [37]. The CASP checklist had ten indicators [35], whereas the CEMB checklist had 11 [36]. The NICE GATE checklist uses a grading system of ++, +, and − [37]. All the studies were appraised using the CASP checklist grading system of low (<5), moderate (5 to 7), and high (≥8) quality. The rating for the NICE GATE checklist was converted into the CASP overall grading system. A high grade was given for ++, a moderate grade for +, and a low grade for −. This method of appraisal permitted easy comparisons and consistency between results. However, all were graded using the CASP checklist; this may affect scores due to the differing nature of the studies appraised. An example of a completed quality assessment can be seen in Appendix A, and an in-depth risk of bias assessment results can be seen in Appendix A, which indicates what each study addressed and the reasoning behind the grades given for the risk of bias per study.

A grading system was then applied to all studies of low (<5), moderate (5 to 7), and high risk of bias (≥8). The risk of bias assessment was completed independently by AR.

A certainty assessment was conducted. For each barrier, a grade was given for a low level of certainty (1), a moderate level of certainty (2 to 5), and a high level of certainty (≥6) (Appendix A). Levels of certainty were equated to the number of times each barrier was mentioned out of the ten studies. For this review, a barrier would be considered conclusive evidence if there were moderate or high levels of certainty across multiple study designs. We would want to see the same barrier across different study designs to indicate that similar results were replicated.

## 3. Results

### 3.1. Study Selection

A total of 9665 studies were identified from the four databases. After 536 duplicates were removed, 9129 studies were screened by title and abstract. Of these studies, the remaining 115 full texts were assessed. In total, 104 studies did not meet the eligibility criteria at the full-text screening stage; therefore, ten were included in the final analysis. A PRISMA flow diagram represents the study selection process (Figure 1). 

### 3.2. Study Characteristics

A total of ten studies were included in the final review. Of these, five were qualitative studies. Four were quantitative cross-sectional studies; the remaining was a quantitative pre- and post-intervention study. Five qualitative studies using semi-structured interviews or focus groups explored individuals’ motivations and public perceptions of PBDs or meat reduction. In addition, four cross-sectional studies examined dietary intake, perceptions, and attitudes towards the consumption of PBDs. Finally, the pre- and post-intervention study explored the effect of health and environment text messages on dietary beliefs/intentions and outcomes relating to behavioural intake changes, specifically with protein, fruit, and vegetable intake changes. All included one or more barriers to adopting a PBD. Dietary intake patterns, changes, and dietary perceptions were the most common outcomes across the studies. In total, 1740 participants were enrolled, of whom 1246 (72%) were female, with the average age ranging between 20 and 42 years across the studies. There was a variation between 9 and 438 participants per study. The largest population sample was from the quantitative cross-sectional study, which focused on adults’ attitudes and knowledge towards plant-based diets. Four studies were set in Europe, three were from North America, and the remaining three were conducted in Australia/Oceania. All included studies were conducted in an urban setting. All studies were published between 2002 and 2021. Six of these studies (60%) were conducted between 2019 and 2021.

### 3.3. Results of Individual Studies

A detailed summary of individual study characteristics and findings is presented in Table 1 and Appendix A.

### 3.4. Synthesised Results

The barriers identified across the included studies were synthesised into 11 themes, as shown in Table 2. This synthesis was based on overarching concepts derived from 40 separate barriers. These approximately followed the order in which they were extracted from the data.

#### 3.4.1. Financial

The perceived high prices of plant-based foods compared to non-plant-based foods were mentioned in two studies [40,45]. For example, one study conducted in Finland [45] considered the perception of high prices of plant-based foods to be the most relevant and significant barrier in their research (r = 0.31, SD = 1.50, *p* < 0.001). This study used a multiple regression model to identify the perceived effect of barriers on climate-friendly food choices. One of the studies [40] that mentioned high prices also discussed the overall cost concern when reducing meat and introducing plant-based foods in New Zealand.

#### 3.4.2. Lack of Knowledge

One of the main barriers (five out of the ten studies) was a lack of information on what to eat if participants were to adopt a PBD [39,40,42,45,46]. This included both eating out and at home. Furthermore, a need for more familiarity with plant-based meals and food was also stated in four studies [39,40,43,46]. This was mainly concerning meat replacements. These two barriers generally contribute to the need for more knowledge surrounding PBDs.

Participants also needed more nutritional knowledge [38,40,44,47]. Some participants believed that consuming non-plant-based foods would meet all nutritional needs, compared to diets that contained a majority or only plant-based foods in New Zealand [40]. Participants felt meeting nutritional needs through an omnivorous diet would be easier. More knowledge regarding the nutrients that plant-based foods contain compared to non-plant-based foods was needed. For example, participants believed they had more nutritional knowledge of non-plant-based foods like red meat. These were considered to have higher iron and protein content. Participants also lacked the nutritional knowledge to understand how to meet their nutritional needs by consuming plant-based foods or a PBD.

Another study set in Scotland found that participants lacking knowledge of PBDs believed that their eating patterns were trivial in their ability to have an environmental impact [44]. More awareness of how diets and eating habits could impact health and the environment was needed to achieve behaviour change. However, the USA pre- and post-intervention study [43] found that those who received environment-focused text messages had a greater change in dietary beliefs than those who received health-focused text messages. This study used a paired-sample *t*-test analysing the outcome of dietary predictors, intentions, and behaviours pre- and post-interventions. Mean value differences (MD) were used to explore the changes in dietary predictors, intentions, and intakes using pre-baseline dietary intake data and post-intervention data for each intervention group. There was an increase in the perceived benefits of consuming a PBD (MD = 0.27, *p* = 0.011), but this did not increase the introduction of plant-based foods into their diets for either intervention group (health SMS, MD = 0.20, *p* = 0.400, environment SMS, MD = 0.05, *p* = 0.811).

Overall education levels impacted knowledge of PBDs. For example, one study in Australia found that university-educated students were more willing to change their diets than non-university-educated people [42].

Four studies found that the terminology surrounding PBDs needed to be clarified to encourage behaviour change [39,41,44,46]. This included the definition of a PBD and the dietary composition of a PBD. One study using data from Belgium, Denmark, the Netherlands, and Spain aimed to understand the awareness, perceptions, and attitudes of PBDs. This study found that participants required help understanding the differences between vegetarian diets, vegan diets, and PBDs. This was deemed the most significant barrier to adopting a PBD when comparing the perceived appeal of the terms PB and vegan or vegetarian diets among four countries using a logical regression model [39].

#### 3.4.3. Emotional

There was distrust in the food system and suppliers, such as supermarkets, as identified in one study set in Scotland [44]. There were concerns about consuming PB alternatives, such as meat replacements, due to the need for more transparency in the food system. Participants wanted to understand how their food was produced.

Stigma played a role in the dietary transition toward a PBD [39,41,46]. A USA study [46] analysed the stigma attached to vegan diets. Those consuming an omnivorous diet would actively distance themselves socially from those who considered themselves vegan. On the other hand, vegetarians experienced less stigma.

Only one study in the USA mentioned food neophobia or the reluctance to try new foods [43]. This study found that despite the intention of participants to decrease animal-based proteins and increase more plant-based foods such as fruit and veg, no significant actual intake changes in these particular foods occurred. Food neophobia and the reluctance to include/increase plant-based foods, particularly plant protein such as legumes, could cause this lack of intake. They suggested that this may be the cause of not accepting new plant foods into the diet and changing their food habits.

Gender stereotypes were mentioned in two studies, whereby certain eating habits were deemed necessary or important depending on gender. For example, husbands believed incorporating more meat-free meals was not masculine. On the other hand, eating meat, having barbeques, etc., was masculine. Husbands did not believe a meal was one without meat. These opinions were seen in all six focus groups set in New Zealand [40]. This is also related to the fear of judgment from others. Others perceived eating a PBD as violating food norms or socially unacceptable in the USA [46]. Fear of judgement from others could overlap with the social theme. However, this barrier was associated more with the emotional responses created by the social context.

Lastly, powerlessness over food choices was a barrier to adopting a PBD. Four studies mentioned this as a barrier [41,44,45,46]. The feeling of powerlessness was due to a lack of information regarding what to eat or a lack of options or control of plant-based foods. Although there is an overlap with the “lack of knowledge” theme, in these studies, it was characterised as an emotional rather than a practical feeling, indicating that feelings of powerlessness may contribute to a lack of behaviour change regardless of actual knowledge levels.

#### 3.4.4. Health

Health barriers played one of the most important roles in adopting a PBD, as a majority of studies (six) mentioned a health-related barrier. Nutritional deficiencies were a concern across three studies [38,42,47]. This barrier is associated with anxiety and worries about not consuming the nutrients required from a PBD, such as B12, iron, etc., in a study conducted in Sweden [47].

Nutritional intake and requirements were the most prevalent health barriers mentioned across studies [38,40,41,42,45,46]. Participants worried that a PBD would only meet some of the nutritional needs required for a healthy, balanced diet and that all nutritional needs would not be met by a PBD. There was also a concern in two studies that dietary diversity needs would not be met by a PBD compared to an omnivorous one [38,41]. Therefore, this barrier is highly linked to the knowledge barrier. The difference is that the nutritional intake and requirements barrier was not only about knowledge but encompassed beliefs/perceptions, accessibility, and physical effects such as feeling fuller and satisfied. Even if individuals knew the nutritional needs required for a balanced PBD diet, these might only sometimes be met. Consuming meat and dairy was deemed as meeting all nutritional needs compared to reducing or removing these foods to incorporate plant-based foods [38,40]. Nutritional knowledge only equates to an individual obtaining all the required nutrients. This barrier is complex as it encompasses other barriers such as cost, tradition, living situation, etc.

#### 3.4.5. Convenience

Four studies stated that time is an essential component of the convenience of adopting a PBD [38,40,41,45]. A lack of time significantly impacted those making decisions to eat plant-based food (r = 0.64, SD = 1.40, *p* < 0.001); this was identified from a multiple regression analysis examining the effect that perceived barriers could have on climate-friendly choices in Finland [45]. Another study set in New Zealand found that the time it took to create plant-based meals was a barrier to adopting PBDs [40].

Two studies cited food preparation and difficulty creating plant-based meals as barriers [38,40]. It was believed that preparing a vegetarian meal would require too much time, practice, and difficulty.

Only one New Zealand study mentioned the increased energy and overall effort needed for this dietary transition towards a PBD [40]. This was described as participants needing to be more conscious about their meat consumption. Understanding food processes would require more effort and mental energy than continuing to eat the majority of non-plant-based foods.

#### 3.4.6. Social

The most common social barrier was the fear of missing out or not following the social norms of society. There was a significant fear of missing out and wanting to eat the same foods as others around them (r = 0.20, SD = 1.59, *p* < 0.001) found in a multiple regression model exploring predictors of choosing climate-friendly foods conducted in Finland [45]. A study in the USA found that individuals who follow a PBD may have to refuse food in social situations, which was considered socially unacceptable by some participants [46]. Furthermore, in a Canadian study, other expectations, not only societal but also from family and friends, influenced decisions to adopt PBDs [38]. Participants felt it would be difficult to justify their eating habits to others in social situations. They did not want others to perceive them as enforcing their beliefs onto others, as found in a study from Sweden [47]. A study conducted in New Zealand found that this was worsened if the participants lived with individuals who did not approve of or consume a PBD [40].

A lack of support/support network in transitioning towards or eating a PBD was a barrier in the two studies. One study in the USA found that having a support network or supportive family could help participants move towards a PBD and reduce this barrier [46].

Finally, one study in Scotland stated that tradition and individual social roles played a key part in preventing meat reduction. In this study, meat, in particular, was considered a traditional way of eating. This was an important factor in dietary choice despite any perceived benefits that may occur from adopting a PBD, including any positive impacts on the environment from reducing meat consumption and increasing more plant-based foods like fruit and vegetables into the diet [44]. This perceived idea of what a traditional way to eat is would prevent individuals from transitioning.

#### 3.4.7. Enjoyment of Meat

Three studies cited personal and family enjoyment of meat as influencing consumption. Family expectations and living situations would affect personal enjoyment of meat due to added pressures or encouragement to consume non-plant-based foods. Resisting temptations were also mentioned as a barrier in one study set in the USA [46]. It was difficult for participants to resist non-plant-based foods, and this prevented them from adopting a PBD.

The taste was noted in three studies [38,41,42], and the craving barrier was also noted in three studies [40,41,44]. The overall enjoyment/taste of meat made participants crave and want to consume meat. Non-plant-based foods were craved less, and this could result in a lack of desire to adopt a PBD in this New Zealand study [41]. In addition, the cravings for meat were noted to increase after physical activity in some participants.

#### 3.4.8. Environmental

Only two studies reported environmental barriers [44,45]. These barriers included scepticism around the dietary impact of non-plant-based foods and disbelief in climate change. Participants felt their dietary patterns had no or limited impact on the environment. Some participants were generally sceptical of a PBD, particularly towards environmental benefits. In the Scotland study, others did not believe in climate change or that their diets would impact climate change [44].

#### 3.4.9. Accessibility

In a study conducted in New Zealand, there were concerns about the lack of PB options, especially when eating out, and the lack of opportunities to buy plant-based foods. This referred to eating out at restaurants or buying plant-based foods to consume/make meals, such as PB cheese [41]. This lack of options was especially related to meat-free options.

The most common barrier in this category was the lack of autonomy and opportunity over food purchases [38,41,42,47]. Although this barrier was seen mainly in participants aged 18–35 who were at the age of being able to buy their own food, participants felt they needed more autonomy over food choices. This was for multiple reasons, but mainly that they lived with their families, and their parents/carers were the individuals who conducted the food shopping in their household.

#### 3.4.10. Personal Ability

A lack of confidence in cooking ability and preparation of plant-based meals was a barrier for one study conducted in New Zealand [41]. Participants in a Canadian study lacked confidence in their cooking ability, particularly in plant-based foods that included lentils, legumes, and tofu. Furthermore, participants’ actual (rather than perceived) cooking ability was also stated as a barrier in two studies [38].

The habit was the second most common barrier across studies. In one Swedish study, men perceived their habitual diet as more important than women’ s. Another study [47] reported that participants had anxieties about exploring new ways of living. Therefore, habitual eating behaviours would reduce anxiety, particularly in combination with fears around nutrient deficiencies with eating a PBD.

#### 3.4.11. Media

The last barrier was media and advertisements promoting meat consumption. However, this was only reported by one study conducted in New Zealand [40]. One participant had experienced advertisements promoting the idea of “meat and three veg” growing up (Figure 2).

A COM-B model framework for synthesised barriers was created. Each barrier was categorised into the COM-B model. As shown in Figure 3, the results of this synthesis show that most barriers were associated with the opportunity section of the model, particularly social opportunity. However, some barriers did overlap in the model, such as a need for familiarity with plant-based foods and PBDs. Individuals may need to gain knowledge of plant-based foods/diets (psychological capability) and resources, such as having supermarkets nearby that sell plant-based foods (physical opportunity). Individuals may also lack familiarity with plant-based foods/diets because they are not commonly eaten in their social circle (social opportunity). From this synthesis, adopting a PBD would encompass multiple components of the COM-B model.

### 3.5. Quality Appraisal

#### 3.5.1. Risk of Bias Assessment Results

For the risk of bias assessment, five studies were considered high quality, four were considered moderate quality, and one was considered low quality (Table 3). Of these synthesised results, five studies had a low risk [39,40,41,42,44] of bias, four had a moderate risk [38,43,46,47], and only one study had a high risk [44] due to sample size considerations, a satisfactory response rate not being achieved, and issues with confounding (not adjusted for or analysed) [45].

#### 3.5.2. Certainty Assessment Results

Nutritional intake/requirement had the highest level of certainty, as six studies had cited this as a barrier across the ten total studies. This indicates that from the evidence available, nutritional intake requirements are the most cited factor for individuals when considering adopting a PBD. Overall, 29 out of the 40 barriers were graded with moderate certainty. Although the craving barrier was at a moderate level, it was cited only by qualitative studies. There was a low level of certainty for 9 out of the 40 barriers (score of 1). For example, disbelief in climate change was of a low level of certainty. This could be because adopting a PBD to reduce climate change could be a facilitator rather than a barrier. Below is the synthesised certainty assessment of all 40 barriers (Table 4 and Appendix A).

## 4. Discussion

### 4.1. Summary of Key Findings

We found a range of studies conducted in HICs that examined the barriers associated with adopting a PBD. Outcome measures varied across studies, with some studies not having defined outcomes. Defined outcomes included dietary intake and change, perceptions, and attitudes. All studies reported one or multiple barriers to adopting a PBD. However, these varied in importance across studies. For example, the most reported across studies were habits, lack of knowing what to eat, and nutritional intake/requirements of a PBD. Of these most cited barriers, nutritional intake/requirements had the highest quality of evidence, as this was cited the most in high- and moderate-quality studies. Comparatively, the other two barriers that were most cited came from a slightly lower quality of evidence, as these barriers included evidence from three high, one moderate, and one lower level of study.

Using the COM-B model to synthesise the findings, it is clear that the barriers extend across all components. The majority of barriers fell into the categories of social opportunity and psychological capability. However, there was an overlap between some barriers. For example, a lack of familiarity with plant-based foods and diets was mapped onto the physical and social opportunity category and psychological capability. For individuals to adopt a PBD, these components must be addressed, particularly the barriers that extend across the model. From mapping using the COM-B model, education interventions and communication/marketing policies should be prioritised to help reduce the barriers individuals experience when adopting a PBD.

### 4.2. Interpretation of Results in the Context of Previous Research

The lack of information on the barrier of knowing what to eat was cited by five studies [39,40,42,45,46]. This barrier consisted of individuals lacking knowledge regarding what can be eaten as part of a PBD and what foods could be replaced or used as alternatives to non-plant-based foods. This included the example of reducing meat consumption and occasionally incorporating meat alternatives. This evidence is consistent with previous study findings. A previous review conducted in Australia found that those contemplating or preparing to adopt a PBD scored highest for information barriers [30]. Another study from Australia found that a lack of knowledge regarding vegetarian diets was the most robust barrier for non-vegetarians to make this transition. This barrier focused on not knowing what to eat instead of consuming non-plant-based foods, particularly meat. This was followed by statements that individuals frequently sought more information on eating healthier [48]. Overall, the wider evidence supports this review, which found that the lack of information on what to eat as part of a PBD is an important barrier to preventing this dietary shift. However, in this review, this barrier was not the most important compared to the previously stated literature.

The perception of adopting a PBD was associated with concerns about daily nutritional intake and the requirements needed. This barrier had the most robust evidence in this review, as it was mentioned across six studies. The concern among participants in Canada was that a PBD would not provide the essential nutrients and energy needed to obtain a healthy and balanced diet. In contrast, participants believed consuming non-plant-based foods such as meat or dairy would meet all these needs [38]. One previous study set in Australia found that perceived health concerns with consuming a vegetarian diet would be not obtaining enough protein or iron from a majority of PBD. This barrier was, therefore, a positive predictor of consuming meat [49].

Despite the health benefits of consuming a predominantly PBD, as mentioned in this review, the previous literature found objective evidence of lower nutritional levels of vitamins B12, iron, vitamin D, iodine, and calcium in those consuming a PBD compared to those following an omnivorous diet. This evidence was found in a systematic review from the Netherlands examining nutritional intake and status in adults consuming either a PBD or an omnivorous diet [50]. In contrast, those following an omnivorous diet were considered only at risk of inadequate nutritional intake of certain nutrients such as calcium, fibre, and magnesium. However, in meat eaters, it was found that vitamin D and E nutrient intake was insufficient [50]. These previous findings demonstrate the objective evidence supporting this perceived barrier (nutritional intake/requirements).

In this review, nutritional intake and requirements were the barriers most cited as preventing an individual from adopting a PBD; this was not reflected in the broader literature of research studies. As many other studies stated, the most important barriers were food neophobia, food preparation, lack of information, enjoyment of meat, and a belief that humans are made to eat large amounts of meat [51,52,53,54,55]. Furthermore, other systematic reviews did not report nutritional intake and requirements as a key barrier/factor preventing the adoption of a PBD. Some of these reviews state that enjoyment of meat, difficulties in removing it from the diet, the stigma associated with adopting a PBD, and a lack of cooking ability are some of the main barriers to adopting a PBD [34,56,57]. This review included several more recent studies focusing on nutritional intake and requirements not reflected in earlier reviews.

In this review, habit also played an essential role in preventing individuals from adopting a PBD. In a study set in Finland, there was an association between habitual eating patterns and the difficulty or unwillingness to transition to a PBD because of this barrier. This barrier was particularly seen in males [45]. Furthermore, personal eating habits were also impacted by an unwillingness or inability of family members or those they live with to change their eating patterns in an Australian study [42]. This would become a more critical barrier if individuals lived with those who followed an omnivorous diet or those who deemed habitual eating patterns more important than making a dietary transition to a PBD. Despite this barrier being one of the most important in this review, there was a lack of evidence in the previous literature regarding the importance of habit as a barrier to adopting PBDs. However, one study conducted in Poland found that habit was the main barrier to consuming pulses. This lack of habit was mainly related to the preparation of the pulses [58]. This barrier identified in the study could also relate to the lack of knowledge barrier found in this study. The lack of evidence for this barrier could be because habit was not easily defined as a barrier and due to the complexity of habit as it encompasses many barriers. For example, habit plays a role in cooking or shopping habits (cooking ability and no autonomy/opportunity over food purchase barriers), eating patterns (personal and family enjoyment of meat, traditions barriers), etc. 

Previous studies set in the UK and a mixed American and Denmark study, respectively, had suggested that the prices of a PBD were the most important barrier to adopting a PBD [27,59]. In a previous review [34], the cost of plant-based diets and foods was mentioned four times as a barrier. Another review using data from ten European countries found that prices, particularly when eating out, were the most important barrier affecting dietary shift with the belief that plant-based food products were too expensive [55]. However, this was found not to be the case in this review. This barrier was on the lower end, as only two studies (moderate level of certainty) suggested this would be a barrier to adopting a PBD [40,45]. Other studies have also mentioned cost as a barrier. However, these studies were not included in this review for numerous reasons. For example, due to the date the study was conducted, one of the studies that looked at attitudes towards meat eating in both vegetarian and non-vegetarians in England was conducted in 1998 [60]. Another study set in 1998 from America found that cost had the second most impact on food choices after taste [61]. Furthermore, studies from low- and middle-income countries were not included in this review; therefore, studies set in these countries with cost as a barrier were omitted. An example of this is an American study that analysed the affordability of the EAT-Lancet diet using retail prices; this included data from 159 countries [62]. The affordability of this diet was possible for high-income countries; however, low-income countries could not afford this shift towards introducing whole-food, plant-based foods. For 26 low-income countries in this study, eating the least expensive part of the EAT-Lancet diet would comprise 89.1% of the household income. These differences could also be seen in the differing country food pricing of plant-based foods.

Furthermore, in this review, environmental barriers were less prevalent than other barriers. There was also a low level of certainty for the disbelief in the climate change barrier. Previous evidence by the EAT-Lancet Commission has shown that consuming a PBD reduces the dietary impact on climate change, such as the amount of greenhouse gas emissions emitted and the requirements for land and water [63]. As previous evidence has suggested, environmental factors could be considered facilitators of adopting a PBD, as well as barriers. One New Zealand study [41] states that environmental concerns increase the chances of a participant adopting a PBD. Another USA study [43] found that those who received an environmental-focused text message were more likely to increase vegetable intake and view PBDs as more beneficial than those who received a health-focused text message.

Lastly, advertisements promoting meat consumption were considered a barrier. However, this was in only one study conducted in New Zealand [40]; therefore, it had low levels of certainty. This was a barrier as it encouraged the participant to consume a non-PBD and see it as traditional, which was embedded in eating habits. This made it harder to break this barrier. Despite this barrier being less important in this review, the previous literature has shown evidence of the influence of media and advertisements on eating habits, particularly the promotion of meat consumption. Some examples of this association were seen in a Norwegian study, which showed that dissociation was caused by the positive promotion of advertisements regarding meat consumption [64]. Others included influencing meat preferences through advertisements. Participants in the USA study who were exposed to advertisements including meat imagery had an increased desire to consume meat compared to those who saw nonmeat imagery [65]. Therefore, the low level of certainty for the advertisement promoting meat consumption barrier in this review should be taken with caution.

### 4.3. Strengths and Limitations

#### 4.3.1. Strengths

A wide range of barriers was identified from a comprehensive search of four databases. These barriers were identified from 11 HICs and associated with preventing individuals from adopting a PBD. To our knowledge, this is the first review of its kind. The results of this study provide evidence of the growing importance of encouraging and facilitating PBD by combating the barriers. The study findings were mapped against the BCW by categorising the barriers into the COM-B model, a validated and well-established model of behaviour change. A quality and certainty assessment was conducted for all studies and barriers; out of the ten studies, only one was of low quality.

#### 4.3.2. Limitations

This review had some limitations in both the review process and the evidence. Firstly, the review process was completed independently by one researcher. This included data collection, analysis, and quality appraisals. This could have led to certain biases, including selection bias, with the data collected. Another limitation could be the key search terms used during the data collection. These keywords may have been too limited. Therefore, certain studies may have been missed. Furthermore, the studies in this review used multiple study designs, such as cross-sectional, qualitative, and pre- and post-intervention studies. This improved the confidence and strength of the evidence. However, due to the methodological diversity, the heterogeneity of the studies meant conducting a meta-analysis would have been impossible. In addition, the heterogeneity of the outcomes per study made it difficult to compare outcomes. This review may have missed other evidence due to the small sample size; therefore, conclusions are limited. Finally, this review included studies from a range of HICs. These study findings may be generalisable to other HICs but cannot be generalised to LMICSs. Most participants in the included studies were female (approximately 72%), so findings may be less applicable to male participants. 

### 4.4. Recommendations on How to Overcome the Most Important Barriers

The lack of information on what to eat as a barrier was mapped against the capability component and placed into the psychological section. Due to the section this barrier was mapped against, an educational programme would be recommended to improve individuals’ psychological capabilities. These educational programmes could teach individuals what can be eaten as part of a healthy, balanced PBD, including not just novel and processed meat and dairy alternatives but also nutrient-rich plant-based foods such as legumes, nuts, healthy fats, etc. This could develop individuals’ knowledge of what to eat as part of a PBD.

The importance of the nutritional intake/requirements barrier in this review implies a significant need for education surrounding this topic. This barrier was mapped against the physical capability component of the COM-B model. In addition, mass education programmes should be employed as a preventive strategy to inform individuals who want to adopt a PBD on the nutritional requirements and intake needed for a healthy, balanced PBD.

The barrier of habit expanded across numerous components of the COM-B model. This included opportunities, both physical and social, as well as automatic motivations. Habit encompasses physical opportunity. This is because an individual’ s environment can influence and alter habitual behaviour. For example, without the accessibility to plant-based foods sold in local supermarkets (physical opportunity), it may be difficult to encourage habit changes due to the lack of physical opportunity. Furthermore, habit is widely affected by the surrounding social and cultural norms. These norms could make it difficult for individuals to change their eating or shopping habits. Lastly, impulses and desires are part of automatic motivation behaviours. These behaviours, therefore, influence individuals’ habits. Due to the complexity of this barrier, multiple interventions should be considered. Recommendations to overcome this barrier could include environmental restructuring by providing more accessible plant-based foods and meals to the public. This could reduce the burden of physical opportunity. In addition, education using mass media campaigns (communication/marketing policies) could help raise awareness by normalising plant-based food and diets and promoting social change. Finally, an intervention that could help with automatic motivations is environmental restructuring interventions. This would give individuals better access to PB alternatives/replacements, which could diminish some automatic motivations, such as desires for non-plant-based foods.

Context is an important factor to consider when evaluating how to overcome these impactful barriers. This is because the context could greatly impact the effectiveness of the suggested recommendations, for example, country differences, such as variations in food pricing for plant-based foods and cultural differences, as well as population and individual distinctions, which include socioeconomic status, gender differences, etc. In addition, as stated above, the recommendation of environmental restructuring to increase the accessibility of plant-based foods and meals could be impacted by an individual’ s financial status. Although plant-based foods and meals would be increased and more accessible with this recommendation, this does not consider the financial context of all individuals. Therefore, contextual factors are important for each recommendation above. This is due to the influence these factors could have on the success of the recommendation in overcoming these most important barriers.

### 4.5. Implications for Policy

The EAT-Lancet Commission states that there needs to be a global shift to consuming a diet rich in PBFs such as vegetables, nuts, legumes, etc., and reducing red meat by 50%. “A diet rich in PBFs and fewer animal source foods confers both improved health and environmental benefits” [55]. Despite the numerous benefits of adopting a PBD for human and environmental health, individuals are still experiencing barriers preventing them from adopting this diet. This review could further aid the understanding of overcoming these barriers to help more individuals who wish to adopt a PBD. In addition, this review could provide valuable evidence of implementations policymakers could enforce to drive individuals’ behaviour change [32] and encourage PBDs. From this review, policymakers and other actors (businesses, NGOs, etc.) should focus on communication and marketing policies. Focusing on this policy area could address the most critical barriers in this review and reduce their burdens for the public. However, similar to the recommendations to overcome the most important barriers, policymakers and other actors must also consider contextual factors when implementing these recommended interventions and policies. These policies would impact the lives of many individuals with differing circumstances. These differences may include educational background and financial status. Therefore, policymakers and other actors should try encompassing contextual factors to effectively maximise outreach to as many individuals as possible.

## 5. Conclusions

To conclude, this review presents comprehensive and recent evidence of barriers to adopting a PBD in HICs dating from 2002 to 2021. Evidence shows that individuals face multiple barriers when making this dietary shift. There were many results, ranging from a lack of information about knowing what to eat on a PBD to the stigma individuals may receive when adhering to a PBD. This synthesis demonstrated that the barriers with the most evidence that should be prioritised were the need to know what to eat as part of a PBD, habits, and nutritional intake/requirements of a PBD. Using the COM-B model highlights the important policy and intervention areas to consider when encouraging dietary behaviour change toward a PBD. Education and environmental restructuring interventions and communication/marketing policies should be used to overcome many barriers. For example, a mass media campaign to educate the public on what can be eaten as part of a balanced PBD. They also provide these plant-based foods/meals in public facilities such as cafeterias. These implementations would address the physical, social opportunity, and psychological capability aspects of behaviour change. These synthesised results could aid policymakers in targeting each component (such as social opportunity) to reduce each barrier’ s effect on behaviour change (adopting a PBD). To further close the gap in research, it may be useful to review the facilitators of adopting a PBD. This would provide further information on the most effective strategies to encourage/facilitate the adoption of PBDs. Future research could also compare the barriers perceived in different types of plant-based diets, such as vegan, flexitarian, vegetarian, etc., to understand the dietary comparisons and barriers between these diets. 

## Figures and Tables

**Figure 1 nutrients-16-00823-f001:**
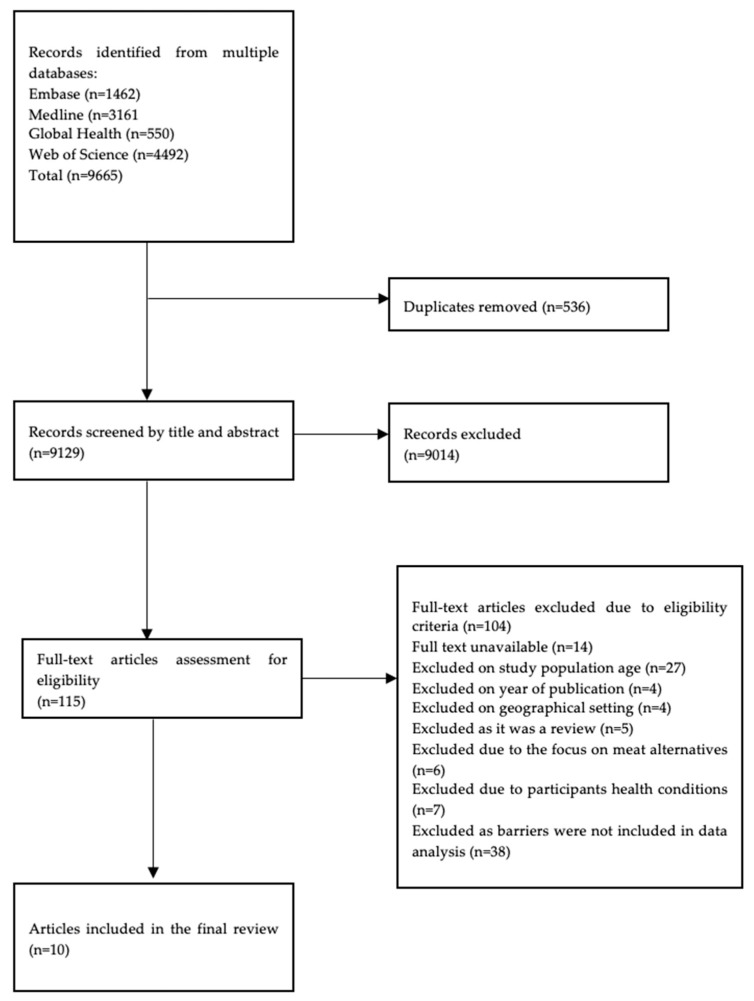
A PRISMA flow diagram of the study selection process for systematic reviews.

**Figure 2 nutrients-16-00823-f002:**
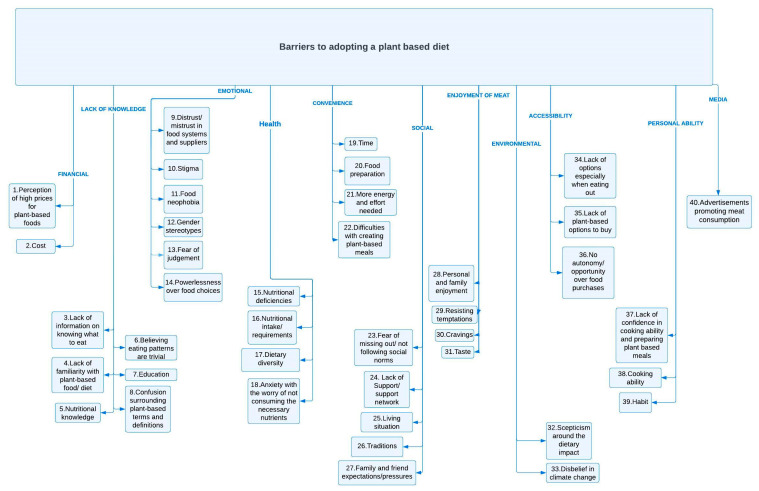
The conceptual map of the visually tabulated synthesised results with links between barriers across the themes.

**Figure 3 nutrients-16-00823-f003:**
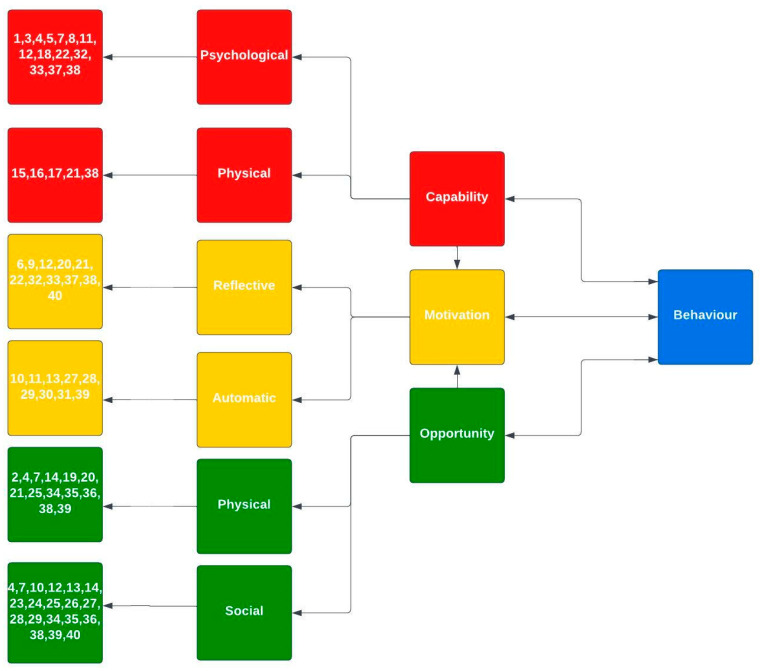
Synthesised barriers in the COM-B model. Numbers represent barriers, as shown in Table 2.

**Table 1 nutrients-16-00823-t001:** Summary of individual study characteristics and findings included in this systematic review (adapted; see full text in Appendix A).

Author, Title, and Year	Study Geographical Setting	Study Population Sample Size, Age, Sex, and Social Position	Study Design	Study Outcome	Data Analysis Method
Barr and Chapman. Perceptions and practices of self-defined current vegetarian, former vegetarian, and non-vegetarian women (2002) [38].	Canada	Total n = 193. Vegetarian n = 90; former vegetarian n = 35; non- vegetarian n = 68. Age: 18–50-year-olds. Females: n = 193. University graduates: 49.7%; students: 40.1%; employed: 73.2%.	Mixed methods. Cross-sectional survey; qualitative interviews with a subsample.	Dietary intake patterns and dietary change; Perception of meat and dairy products.	Group comparisons: One-way ANOVA. Continuous variables: Post hoc testing. Categorical variables: Chi-squared.
Faber, Castellanos-Feijoo, Van de Sompel, Davydova and Perez-Cueto. Attitudes and knowledge towards plant-based diets of young adults across four European countries (exploratory survey) (2020) [39].	Belgium, Denmark, the Netherlands, and Spain	Total n = 438. Belgium n = 110; Denmark n = 119; Netherlands n = 116; Spain n = 93. Age: 18–30 years. Females: 56–82% across countries. Males comprise 18–44% of the population overall across countries.	Quantitative cross-sectional study.	Awareness of plant-based diets- perception and attitudes; The appeal of diet terminology.	Differences among countries: Pearson’ s Chi-squared test. Proportions: Fisher’ s exact test. Continuous variables: Kruskal–Wallis test. To test the association between knowledge and attitudes towards PB, vegan, and vegetarian diet terms. Adjusted for age, sex, etc.: logical regression.
Kemper. Motivations, barriers, and strategies for meat reduction at different family lifecycle stages (2020) [40].	New Zealand	Total n = 36. Age: 18–60 years. Females: n = 32. Males: n = 4.	Qualitative research with the use of focus group interviews.	Motivations; Barriers; Strategies; Meat reduction.	Thematic analysis.
Kemper and White. Young adults’ experiences with flexitarianism: The 4Cs (2021) [41].	New Zealand	Total n = 23. Age: 18–35 years. Females: n = 17. Males: n = 6. Students: 100%.	Qualitative research with an exploratory approach using semi-structured interviews.	Lived experiences; Motivations; Strategies; Barriers; Meat reduction and transition to a flexitarian diet.	Thematic analysis.
Lea, Crawford, and Worsley. Public views of the benefits and barriers to the consumption of a plant-based diet (2006) [42].	Australia	Total n = 415. Age: 20–65 years. Females: 59.4%. Males: 40.6%. Employed full-time: 31.5%; employed part-time: 17.8%; unemployed: 2%.	Quantitative cross-sectional survey study design.	Perception and attitudes towards dietary intake/change; Barriers; Benefits.	Frequency of responses: Pearson’ s Chi-squared test (adjusted by sex, age, and university education).
Lim, Okine, and Kershaw. Health- or environment-focused text messages as a potential strategy to increase plant-based eating among young adults: an exploratory study (2021) [43].	The United States of America	Total n = 159. Age: 18–26 years. Females: n = 107. Males: n = 49. Other: n = 3.	Quantitative 8-week text message pre- and post-intervention.	Dietary beliefs, intentions, behaviour, and intake.	Participant’ s characteristics group comparisons: Chi-squared test. Explore the effect of the interventions on dietary predictors, intentions, and behaviours: Paired samples *t*-test, from baseline and post-intervention answers. Differences between group intervention: Independent samples *t*-test.
Macdiarmid, Douglas, and Campbell. Eating like there’ s no tomorrow: public awareness of the environmental impact of food and reluctance to eat less meat as part of a sustainable diet (2016) [44].	Scotland	Total n = 83. Age: 25–56 years. Females: n = 43. Males: n = 40.	Qualitative research with the use of focus group interviews.	Public awareness; Public willingness to make dietary changes; Meat reduction.	Thematic analysis. Exploring deprivation, sex or urban/rural differences in attitudes towards reducing meat consumption: Framework analysis.
Makiniemi and Vainio. Barriers to climate-friendly food choices among young adults in Finland (2014) [45].	Finland	Total n = 350. Mean age: 24. Females: n = 280. Males: n = 70. Students: 100%.	Quantitative cross-sectional, questionnaire study.	Perception and attitudes, and barriers toward dietary intake/change.	Differences in men and women for perceived barriers: *t*-test. The impact of perceived barriers on climate-friendly food choices: multiple regression analysis. The impact of perceived barriers on climate-friendly food choices: multiple regression analysis.
Markowski and Roxburgh. “If I became a vegan, my family and friends would hate me:” Anticipating vegan stigma as a barrier to plant-based diets (2019) [46].	The United States of America	Total n = 34. Females: n = 26. Males: n = 8. Students: 100%.	Qualitative interviews with the use of focus groups.	Individual perceptions; Stigmatisation; Barriers; Meat consumption.	Coding of transcripts.
Von Essen. Young adults’ transition to a plant-based diet as a psychosomatic process: A psychoanalytically informed perspective (2021) [47].	Sweden	Total n = 9. Age: 18–35 years.	Qualitative semi-structured interviews.	Individual perceptions; Dietary transition; Challenges.	Descriptive phenomenological psychological method.

**Table 2 nutrients-16-00823-t002:** The 40 barriers identified from the thematic analysis of studies included in this review are aggregated by theme.

Barrier Identified	References of the Studies That Included Each Barrier
** *Financial* **	
1. Perception of high prices for plant-based foods;	40, 45
2. Cost.	40
** *Lack of knowledge* **	
3. Lack of information on knowing what to eat;	39, 40, 42, 45, 46
4. Lack of familiarity with plant-based foods/diets;	39, 40, 43, 46
5. Nutritional knowledge;	38, 40, 44, 47
6. Believing eating patterns are trivial;	44
7. Education;	40, 42
8. Confusion surrounding the terms and definitions of plant-based diets.	39, 41, 44, 46
** *Emotional* **	
9. Distrust/mistrust in food systems and suppliers;	40, 44, 45
10. Stigma;	39, 41, 46
11. Food neophobia;	42
12. Gender stereotypes;	40, 42
13. Fear of judgement;	41, 46
14. Powerlessness over food choices.	41, 44, 45, 47
** *Health* **	
15. Nutritional deficiencies;	38, 42, 47
16. Nutritional intake/requirements;	38, 40, 41, 42, 46, 47
17. Dietary diversity;	38, 41
18. Anxiety with the worry of not consuming the necessary nutrients.	38, 47
** *Convenience* **	
19. Time;	38, 40, 41, 45
20. Food preparation;	38, 40
21. More energy and effort needed;	40
22. Difficulties with creating plant-based meals.	38, 40
** *Social* **	
23. Fear of missing out/not following social norms;	41, 45, 46
24. Lack of support/a support network;	38, 46
25. Living situation;	38, 41
26. Traditions;	44
27. Family and friend expectations and pressures.	38, 40, 46
** *Enjoyment of meat* **	
28. Personal and family enjoyment;	38, 39, 40
29. Resisting temptations;	46
30. Cravings;	40, 41, 44
31. Taste.	38, 41, 42
** *Environmental* **	
32. Scepticisms around the dietary impact;	44, 45
33. Disbelief in climate change.	44
** *Accessibility* **	
34. Lack of options, especially when eating out;	41, 44, 46
35. Lack of plant-based options to buy;	41, 45
36. No autonomy/opportunity over food purchases.	38, 41, 42, 47
** *Personal ability* **	
37. Lack of confidence in cooking ability and preparing plant-based meals;	41
38. Cooking ability;	38, 41
39. Habit.	40, 42, 44, 45, 47
** *Media* **	
40. Advertisements promoting meat consumption.	40

**Table 3 nutrients-16-00823-t003:** Results from the risk of bias assessment, including the study used, the score given, and the grading of the score converted to each assessment type.

Study	Score	Grading
Kemper (2020) [40].	9	High
Kemper and White (2021) [41] .	9	High
Macdiarmid, Douglas, and Campbell (2016) [44].	9	High
Markowski and Roxburgh (2019) [46].	7	Moderate
Essen (2021) [47].	7	Moderate
Barr and Chapman (2002) [38].	6	Moderate
Faber, Castellanos-Feijoo, Sompel, Davydova, and Perez-Cueto (2020) [39].	9	High
Lea, Crawford, and Worsley (2006) [42].	8	High
Makiniemi and Vainio (2014) [45].	5	Low
Lim, Okine, and Kershaw (2021) [43].	+	Moderate

**Table 4 nutrients-16-00823-t004:** Certainty assessment results. The first column shows the 40 barriers identified. The second column is the level of certainty. This ranged from low to moderate to high. A low level of certainty is when one study mentioned that barrier. A moderate level of certainty ranged from two to five studies mentioning that particular barrier. Lastly, a high level of certainty included six or more studies showing evidence that a particular barrier prevented individuals from adopting a PBD. Every barrier was mentioned at least once; the most a barrier was mentioned was six times.

Barriers Identified	Level of Certainty	Times Mentioned
1. Perception of high prices for plant-based foods	Moderate	40, 45
2. Cost	Low	40
3. Lack of information on knowing what to eat	Moderate	39, 40, 42, 45, 46
4. Lack of familiarity with plant-based foods/diets	Moderate	39, 40, 43, 46
5. Nutritional knowledge	Moderate	38, 40, 44, 47
6. Believing eating patterns are trivial	Low	44
7. Education	Moderate	40, 42
8. Confusion surrounding the term and definitions of plant-based diets	Moderate	39, 41, 44, 46
9. Distrust/mistrust in food systems and suppliers	Moderate	40, 44, 45
10. Stigma	Moderate	39, 41, 46
11. Food neophobia	Low	42
12. Gender stereotypes	Moderate	40, 42
13. Fear of judgement	Moderate	41, 46
14. Powerlessness over food choices	Moderate	41, 44, 45, 47
15. Nutritional deficiencies	Moderate	38, 42, 47
16. Nutritional intake/requirements	High	38, 40, 41, 42, 46, 47
17. Dietary diversity	Moderate	38, 41
18. Anxiety with the worry of not consuming the necessary nutrients	Moderate	38, 47
19. Time	Moderate	38, 40, 41, 45
20. Food preparation	Moderate	38, 40
21. More energy and effort needed	Low	40
22. Difficulties with creating plant-based meals	Moderate	38, 40
23. Fear of missing out/not following social norms	Moderate	41, 45, 46
24. Lack of support/a support network	Moderate	38, 46
25. Living situation	Moderate	38, 41
26. Traditions	Low	44
27. Family and friend expectations and pressures	Moderate	38, 40, 46
28. Personal and family enjoyment	Moderate	38, 39, 40
29. Resisting temptations	Low	46
30. Cravings	Moderate	40, 41, 44
31. Taste	Moderate	38, 41, 42
32. Scepticisms around the dietary impact	Moderate	44, 45
33. Disbelief in climate change	Low	44
34. Lack of options, especially when eating out	Moderate	41, 44, 46
35. Lack of plant-based options to buy	Moderate	41, 45
36. No autonomy/opportunity over food purchases	Moderate	38, 41, 42, 47
37. Lack of confidence in cooking ability and preparing plant-based meals	Low	41
38. Cooking ability	Moderate	38, 41
39. Habit	Moderate	40, 42, 44, 45, 47
40. Advertisements promoting meat consumption	Low	40

## Data Availability

Data are contained within the article and Appendix A. The data presented in this study are available in Appendix A.

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
