# Peer review of "Barriers to Adopting a Plant-Based Diet in High-Income Countries: A Systematic Review"

_nutrients, 2024, doi:10.3390/nu16060823_

Round 1

Reviewer 1 Report

Comments and Suggestions for Authors

The authors have focused the introduction appropriately. The methodology is well designed, both the analysis of qualitative studies, as well as mixed and studies with quantitative variables. However, it is very difficult to read the whole article and make an assessment beyond the written text because the tables and figures cannot be read properly.

The question to be solved in order to publish this article is the format of the tables and figures.

Table 1 with horizontal format and page headers on all pages;

Table 2: adjust page legend to table format.

Figure 2 is unreadable, it should be in horizontal format on a single page and large enough to be readable.

The legend of figure 3 should be modified because the numbers refer to the studies that reflect the barriers, not the barriers themselves.

"Figure 3. Synthesised barriers in the COM-B model. Numbers represent barriers, as seen in Table 2". Modify as previous comment.

Table 3. Adjust table legend to table format.

Figure 4: nothing can be read, nor can it be understood what it means. Change formatting, increase font size, etc.

Abstract: Line 18-19: The sentence about the results should be clearer about the findings. 

"The barriers with the most evidence were knowing what to eat as part of a PBD, habits and nutritional intake/ requirements." The text part: "habits and nutritional intake/ requirements", it is not very clear what is being referred to. Rewrite the sentence listing all the perceived barriers, emphasising the two with the most evidence found.

Reviewer 2 Report

Comments and Suggestions for Authors

 Dear Authors,

The aim of the submitted manuscript entitled “Barriers to adopting a plant-based diet in high-income countries: a systematic review” was to synthesize the evidence published between  2000 and 2022 on the barriers that adults living in selected (HI) countries experience when adopting a PBD. The barriers were mapped to behaviour change strategies using the COM-B model which is in the centre of the behaviour change wheel (BCW). Most of the barriers were associated with the opportunity section of the model, particularly social opportunity.

Although the number of studies included in the final review was rather small (10) the Authors  identified more than 40 barriers which were synthesized into 11 themes. The Discussion includes an interesting summary of the findings, interpretation of results, strengths and weaknesses, recommendations as well as implications for policy.

In an attempt to improve the article I would like to point out the following:

1The terms “sustainable diets” and “sustainable development goals” are listed as Key words however appear in the manuscript only in the References. I totally agree that PBDs are very important in increasing the sustainability of food systems and achieving the SDGs therefore in this context it  would be important to add a paragraph (in the paper’s Introduction) on SDGs (especially SDG12) and elaborate on the term “plant-based diet” as an element of a healthy and sustainable diets which is a wider concept (ca6640en.pdf (fao.org).

2L93-L99  the cited study did not include only Hungarian studies and papers published up to 1974. Please check and correct this paragraph.

3L225-226 – please specify “last five years” (which years?), were all the studies published between 2002-2021? Also please clarify if all studies were conducted among urban consumers, if not which studies included rural inhabitants.

4 When reading the article it becomes clear that a key term that appears very frequently in the manuscript (42 times) is plant-based foods”. I would suggest clarifying that this term does not include only processed foods (such as meat alternatives) but first of all natural, rich in nutrients foods such as legumes, nuts and seeds (this may be explained in many parts of the paper, also L638 – what can be eaten as part of a PBD).

Although it was mentioned in L691 that “studies focusing on meat alternatives were excluded from the review” (I agree this was a good decision) it seems that many of the analysed studies de facto identified many barriers linked to reducing the consumption of meat and dairy products. An interesting article on barriers to increasing pulse consumption (which is very low in many HICs) which could also add to the paper’s discussion part 4.2 (for some reason did not make it through the selection process) was published in Nutrients in 2020 Nutrients | Free Full-Text | Towards More Sustainable Diets—Attitudes, Opportunities and Barriers to Fostering Pulse Consumption in Polish Cities (mdpi.com)

 L690 is not clear -  it somehow implies that “accessibility to plant based-foods” is  a behavioural change. How is this an implication for policy? Is this a reference to urban food desserts? Scientific papers also impact food and nutrition policy tools A review on policy instruments for sustainable food consumption - ScienceDirect

7      L699 – please specify the years – for those reading this paper in the future.

8      L702  “becoming PB” is probably a colloquial, shortcut expression that many non-native English language Readers may not understand. When can a consumer say he is “PB”?

L757 EAT-Lancet (capital letters)

Comments on the Quality of English Language

needs minor editing.

Reviewer 3 Report

Comments and Suggestions for Authors

The study is interesting and well designed. I suggest to explain why you excluded vegan diets and why your review did not include middle income countryes ( I understand why low income are excluded, but limiting the study to HIC introduces quite a big bias in conclusions)

Reviewer 4 Report

Comments and Suggestions for Authors

Dear Authors,

The reviewed manuscript is of high scientific quality. I think that readers around the world will read it with interest and satisfaction, as was my case. Why do I think so? Firstly, identifying barriers to the transition to PBD in HICs is of great importance in creating and using food policy tools for human and planetary health. Secondly, in this systematic review of the literature, the authors searched four databases and found a methodical key to show the identified barriers (i.e. Behaviour Change Wheel and COM-B model). Thirdly, all parts of the manuscript are written in a clear manner and present the right content. I particularly appreciate the detailed presentation of the material and methodology. Fourthly, the final section of the manuscript includes applied recommendations for mitigating the impact of the identified barriers, including policy implications (food and nutrition, education, social).

In the conclusion of my review, I would like to draw the authors' attention to a few minor points.

L. 28. I am aware of the indicated reference and do not recall it stating that "130 countries....". I would ask the authors to check this information.

L. 128. In the section indicating the publication period of the analysed articles, it is worth repeating the cut-off point (May 2022).

Figure 1. In the last field on the right (exclusion for health reasons), the number 'n' is not shown.

Figure 2. Only for some barriers are the reference numbers indicated. In my opinion, there is no need to enter them in this diagram as the reference numbers are given in Table 2. 

Figure 3. In the COM-B model the lines connecting C, M and O with Behaviour are ended with arrows on both sides.

L. 570. Is the sentence "Or they were not on the databases I had searched [54]" well written here?

L. 619 and 621. In my opinion, these are not limitations of the study, just further issues worth investigating to have a complete picture of the barriers. Instead, the selection of keywords to search for publications in databases may be a limitation.

L. 719. In LICs, the average diet is plant-based, so looking for barriers in these countries misses the point. Especially since an improvement in the economic situation of the population always results in an increase in meat consumption.

It was indeed a pleasure to review this manuscript.

Yours sincerely
